# Clinical laboratory hematology reference values among infants aged 1month to 17 months in Kombewa Sub-County, Kisumu: A cross sectional study of rural population in Western Kenya

Jew Ochola Ouma[1,2]*, David H. Mulama[2], Lucas Otieno[1], John Owuoth[1], Bernhards Ogutu[1], Janet Oyieko[1], Jackson C. Korir[2], Peter Sifuna[1], Valentine Singoei[1], Victorine Owira[3], Stacey Maureen Okallo Gondii[1], Ben Andagalu[1], Walter Otieno[1]

1 KEMRI/USAMRD-K, Walter Reed Project, Kisumu, Kenya, 2 Department of Biological Sciences, Masinde Muliro University of Science and Technology, Kakamega, Kenya, 3 Africa Clinical Research Management, Nairobi, Kenya

* ochojew@yahoo.com, jew.ouma@gmail.com

## Abstract

There is an urgent need for reliable region-specific hematological reference values for clinical monitoring. Laboratory reference ranges are important for assessing study participant eligibility, toxicity grading and management of adverse events in clinical trials and clinical diagnosis. Most clinical laboratories in Kenya rely on hematological reference values provided by instrument manufacturers and/or textbooks, which are based on population from Europe or North America. The use of such values in medical practice could result in improper patient management, selection bias in selection of appropriate participants for clinical trials and flawed classification of the clinical adverse events when applied to African populations. The aim of this study was to establish local laboratory hematological reference values in infants aged 1 month to 17 months from Kombewa Sub-county that could be true representative of the existing rural population. The study participants in the current study were those who had previously been recruited from GSK-sponsored study. This study was a phase III, Double Blind, Randomized, GSK-sponsored, Malaria Vaccine Clinical Trial that was conducted in infants aged 1month to 17months. 1,509 participants were included in the study analysis. Data were partitioned into 3 different age groups (1–6 months[m], 6–12 m and 12–17 m) and differences between gender were compared within each group. Data were analyzed using Graphpad prism V5 to generate 95% reference ranges (2.5th-97.5th percentile). There was evidence of gender differences in hemoglobin values (p = 0.0189) and platelet counts (p = 0.0005) in the 1 to 6m group. For the 12-17m group, there were differences in MCV (p<0.0001) and MCH (p = 0.0003). Comparing gender differences for all age groups, differences were noted in percent lymphocytes (p = 0.0396), percent monocytes (p = 0.0479), percent granulocytes (p = 0.0044), hemoglobin (p = 0.0204), hematocrit (p = 0.0448), MCV (p = 0.0092), MCH (p = 0.0089), MCHC (p = 0.0336) and absolute granulocytes (p = 0.0237). In 1 to 6m age group and all age groups assessed, for WBCs,

**Data Availability Statement:** All relevant data are within the manuscript and its Supporting Information files.

**Funding:** The organization that supported the study is KEMRI/USAMRD-K, Walter Reed Project, Kisumu, Kenya. The funders had no role in study design, data collection and analysis, decision to publish, or preparation of the manuscript. No author received salary from the funders, as the funders had no role in the study. VO is an employee of Africa Clinical Research Management but this company did not have any additional role in the study design, data collection and analysis, decision to publish, or preparation of the manuscript. The specific roles of this author are articulated in the 'author contributions' section. This does not alter our adherence to PLOS ONE policies on sharing data and materials.

**Competing interests:** Author VO is an employee of Africa Clinical Research Management but this company did not have any additional role in the study design, data collection and analysis, decision to publish, or preparation of the manuscript. The specific roles of this author are articulated in the 'author contributions' section. This does not alter our adherence to PLOS ONE policies on sharing data and materials.

hemoglobin, hematocrit, MCV and lymphocytes absolute counts, both $2.5^{th}$ and $97.5^{th}$ percentiles for Kisumu infants were higher than those from Kilifi. Platelet ranges for Kisumu children were narrower compared to Kilifi ranges. Kisumu hematology reference ranges were observed to be higher than the ranges of Tanzanian children for the WBCs, absolute lymphocyte and monocyte counts, hemoglobin, hematocrit and MCV. Higher ranges of WBCs, absolute lymphocyte and monocyte counts were observed compared to the values in US/ Europe. Wider ranges were observed in hemoglobin, hematocrit, and MCV. Wider ranges were observed in platelet counts in Kisumu infants compared to the US/Europe ranges. Compared to Harriet Lane Handbook reference values that are used in the area, higher counts were observed in WBC counts, both absolute and percent lymphocyte counts, as well as monocyte counts for current study. Wider ranges were observed in RBC, platelets and RDW, while lower ranges noted in the current study for hemoglobin, hematocrit and granulocyte counts. This study underscores the importance of using locally established hematology reference ranges of different age groups in support of proper patient management and for clinical trials.

## Introduction

Locally derived reference values for hematological indices are urgently needed to account for regional differences in ecology, ethnicity, nutrition, race, gender among many other variables [1, 2]. These values are important for study participant selection and screening, diagnosis as well as proper patient management [1–5]. Most of the interpretation of hematological indices currently used in many African countries is drawn from values based on exotic population in Europe or North America. Reliable clinical laboratory reference values are critical component in comparative decision-making process for making a medical diagnosis and other physiological assessment [3, 4]. Established values provide an important tool for patient management and influences decisions on participant inclusions or exclusion in clinical studies [5]. These values are important for properly screening of study participants into clinical research studies, monitoring patho-physiological changes after infection or disease states, or following the administration of drugs in therapeutic or clinical interventions and vaccine studies [6].

Research studies are increasingly being carried out in Africa, especially preventive intervention trials for infectious diseases. Although numerous steps have been put in place with the aim of improving the research infrastructure worldwide, laboratory reference ranges used for trial screening and evaluating adverse events are often based on data derived predominantly from European and American populations [7]. Reference interval values are classically derived from biometric parameters that fall within two standard deviations (95% Confidence Interval) of the mean of a healthy population. It is a widely accepted principle that global populations should establish and validate reference values based on the prevailing circumstances and conditions.

Hematological reference values for residents of Kombewa Sub-County have never been established. The values, which are currently used for the local population, are adopted from textbooks and those provided by the analyzer's manufacturer. Previous reference range reports from Africa have shown differences to European and North American populations [5, 8]. The current in clinical trials in Africa and Kenya in particular, has heightened the need for regional and locally established reference values. It is a well-established fact that reference values vary

considerably in different populations, geographic regions, climate, and race and due to other factors e.g. gender, age, genetical factors as well as dietary patterns [8, 9]. The objective of this study was to establish locally derived reference values that are age specific for infants aged 1 month to 17 months in Kombewa Sub-County of Kisumu County, Kenya. The study participants were predominantly from a rural community. The study was undertaken in a cohort of children who had participated in preventive malaria vaccine trial.

## Methods

### Study site

The study was conducted in Kombewa Sub-Country, which is part of Kisumu County, Kenya. The study site lies at 34˚45’ E 0˚10’ S, at an average elevation 1,289 m above sea level, near the northeast shore of Lake Victoria and 40 km northwest of the County capital of Kisumu town. This is a predominantly rural population and mainly of the Luo ethnic group. The stable diet is maize meal, sorghum, millet, cassava, fish and local vegetables. The region is a malaria holo-endemic region as well where other parasitic diseases such as schistosomiasis are prevalent.

### Study design

This was a cross sectional study nested with a phase III GSK-sponsored study. Data used in the study were based on laboratory generated values from a larger phase III, double-blinded, malaria Vaccine Clinical Trial (clinical trial registration number NCT00866619).

### Study population and participant recruitment

The Kombewa Center operates a Health and Demographic Surveillance System (HDSS) which conducts biannual demographic and syndromic surveillance surveys in addition to collecting population data (including in- and out-migrations, births, deaths and verbal autopsies to assign cause of death) for participating households. The HDSS study area population is approximately 150,000 over a 369 square km area and serves as the catchment area for most epidemiologic and research studies conducted at the CRC in Kombewa. This study area has been cartographically mapped by the KEMRI/WRP Kombewa HDSS (KHDSS) using Global Positioning System (GPS) technology. Every building within the HDSS area has been marked and, therefore has an address. The KHDSS is a longitudinal population registration system designed to track the evolving demographic and health status of potential study participants over time. Various studies using this platform take advantage of the sampling frame inherent in the HDSS, whether at individual, household/compound or regional levels. Participants enrolled in the primary study were linked to the HDSS and as such the sample and thus the data for the reference range was a true representative of the population. This sub-study used the data that were generated under a GSK-sponsored Vaccine Clinical trial that was conducted in infants and children between August 2009 and January 2014 [10, 11].

A qualified physician attached to the project performed physical assessment of all the participants and only healthy participants were recruited. Data from normal physical examination were extracted from the files as per the inclusion criteria only to include healthy infants without acute or chronic, clinically significant pulmonary, cardiovascular, hepatobiliary, gastrointestinal, renal, neurological, mental or hematological functional abnormality or illness that required medical therapy. This was as determined through **medical history, physical examination** (blood pressure, weight, pulse, Z score and vital signs) or clinical assessment before being enrolled into the study. Data from children, who had recurrent infection, fever, including HIV and malaria or severe anemia, defined as hemoglobin level of <5.0g/dl or hemoglobin

concentration of <8.0g/dl associated with clinical signs of heart failure and/or severe respiratory distress or those receiving medical treatment at the time of sample collection were excluded from the study.

## Ethical considerations and approval

This study was approved by the Kenya Medical Research Institute Ethics Review Committee and the Walter Reed Army Institute of Research (WRAIR) Institutional Review Boards. The study was conducted in accordance with Good Clinical Practice (GCP), the Declaration of Helsinki and local national regulations of the Kenya Government Expert Committee on Clinical Trials of the Pharmacy and Poisons Board. Approval from the study sponsor, the Malaria Vaccine Initiative, was sought and granted before the reference range study was performed. Written and signed informed consent was sought from study participants before samples were drawn.

## Disclaimer

The investigators have adhered to the policies for protection of human subjects as prescribed in AR 70–25.

## Blood collection, hematological analysis and quality control

Whole blood was collected by phlebotomy in 0.5ml microtainer tubes containing ethylene diamine tetra acetic acid (EDTA) (Becton Dickinson, Franklin Lakes, NJ). Hematology analysis was performed within 24 hours of specimen collection using a three-Part differential Coulter counter hematology analyzer. Prior to use, the hematology instrument was validated on site using validation procedures from Contract Laboratory Services (CLS), South Africa. CLS also conducted Good Clinical Laboratory Practices (GCLP) training to the laboratory staff who were involved in the study. Internal quality control samples with known concentrations had to be analyzed on the hematology equipment and results verified to be within CLS provided ranges before patient samples could be analyzed. The laboratory also registered in a UKNEQAS External Quality assurance provider, which provides two EQA samples per month.

White blood cells (WBCs), red blood cells (RBCs), hemoglobin (HGB), hematocrit (HCT), platelets, granulocytes, lymphocytes, and monocytes were directly measured. Red blood cell indices of MCV (fl), MCH (pg), and MCHC (g/dL), RDW and MPV were extrapolated. The analyzer did not give separate counts of neutrophils, basophils and eosinophils.

## Data management and statistical analysis

Data were verified by the Principal Investigator as per the inclusion and exclusion criteria from participants' folders as part of quality control check. The data is available for access.

The study participants were stratified by age and gender into the following predetermined groups: 1 to 6 months, 6 to 12 months and 12 to 17 months. Both the 95% reference values and 90% CI limits were calculated. Results were evaluated for a normal distribution using the Kolmogorov-Smirnov and Shapiro-Wilk tests. When normality could not be achieved, non-parametric methods were used and the 2.5th and 97.5th percentiles were estimated to for the 95% reference interval. Data were analyzed using Prism V8 (SanDiego, USA). A two-sided p value of <0.05 was considered significant. Reference intervals were considered the central 95% interval of 2.5th and 97.5th percentile as per the CLSI 2008 guidelines [3]. All of the information/data was fully anonymized prior to access.

**Table 1. Proportion of males and females and the median age for the different age categories.**

| Age groups (Months) | Males (%) | Females (%) | Total (%) | Median age [months] (range) |
|---|---|---|---|---|
| 1 to 6 | 362 (24) | 394 (26) | 756(50.1) | 1 (1 to 6) |
| 6 to 12 | 200(13) | 187 (12) | 387(25.6) | 10 (6 to 12) |
| 12 to 17 | 149 (10) | 217 (14) | 366(24.3) | 15 (12 to 17) |
| Total | 711(47.1) | 798(52.9) | 1509(100) | |

Table 1 shows the proportions of males and female participants in the study and the median age for the various age categories. (Numbers in parenthesis show the percent proportion of each category for each gender and the total as well as the ranges for the age categories).

## Results

This study used baseline laboratory results for a 1,509 enrolled participants enrolled in the malaria vaccine trial that enrolled 1631 children aged 1 to 17 months. Reasons for exclusion included: Sickle cell anemia (N = 2; 0.11%), fever (N = 10; 0.57%), HIV exposed (N = 101, 5.76%) and other acute illnesses (N = 9; 0.51%). The majority of infants (50.1%) were in the 1m to 6m age range and overall there were marginally more males than females (Table 1). The median age was 6 (IQR 1.25–12) months (Table 1).

Fig 1 shows the flow diagram on recruitment of study participants for the reference range analysis.

Table 2 shows 95% hematology reference ranges for the different analytes in children aged 1 month to 17months from Kisumu as derived in this study.

### Erythrocyte indices vary according to gender in children in Kisumu County

To establish the differences in the median values for hemoglobin, hematocrit, RBCs, MCH, MCHC and MCV in regard to gender, median values were compared across the study groups.

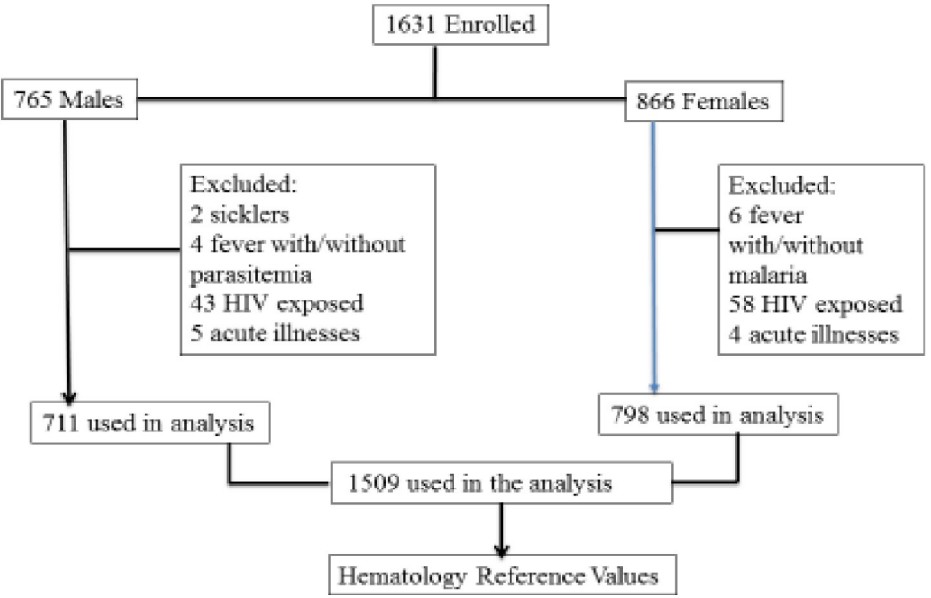

**Fig 1. Flow diagram on recruitment of study participants.**

**Table 2. 95% hematology reference ranges for hematology analytes in infants aged 1-17 months from Kisumu.**

| Parameter/age group | | N | Median | 95% reference values | 90% Confidence Interval (Lower Limits) | 90% Confidence Interval (Higher Limits) |
|---|---|---|---|---|---|---|
| **(A) (White blood cells and percent lymphocytes)** | | | | | | |
| **White Blood Cells (×10³ cells/µl)** | | | | | | |
| 1 to 6 months | Male | 362 | 8.9 | 5.2–19.5 | 4.7–5.7 | 19.0–20.0 |
| | Female | 394 | 9.2 | 5.0–16.9 | 4.5–5.5 | 16.4–17.4 |
| 6 to 12 months | Male | 200 | 11.2 | 6.2–19.8 | 5.7–6.7 | 19.3–20.3 |
| | Female | 187 | 12.0 | 6.5–22.1 | 6.0–7.0 | 21.6–22.6 |
| 12 to 17 months | Male | 149 | 11.2 | 6.5–21.3 | 6.0–7.0 | 20.8–21.8 |
| | Female | 217 | 10.9 | 6.3–19.5 | 5.8–6.8 | 19.0–20.0 |
| Overall | Male | 711 | 10.0 | 5.8–20.0 | 5.3–6.3 | 19.5–20.5 |
| | Female | 798 | 10.1 | 5.6–19.3 | 5.1–6.1 | 18.8–19.8 |
| Overall | All gender | 1509 | 10.1 | 5.7–19.8 | 5.2–6.2 | 19.3–20.3 |
| **Percent Lymphocytes (%)** | | | | | | |
| 1 to 6 months | Male | 362 | 70.0 | 52.5–83.6 | 52.0–53.0 | 83.1–84.1 |
| | Female | 394 | 69.4 | 49.1–81.8 | 48.6–49.6 | 81.3–82.3 |
| 6 to 12 months | Male | 200 | 69.8 | 50.7–82.3 | 50.2–51.2 | 81.3–82.3 |
| | Female | 187 | 70.1 | 48.5–82.5 | 40.0–49.0 | 82.0–83.0 |
| 12 to 17 months | Male | 149 | 67.7 | 48.0–81.8 | 47.5–48.5 | 81.3–82.3 |
| | Female | 217 | 66.4 | 44.8–78.5 | 44.3–45.3 | 78.0–79.0 |
| Overall | Male | 711 | 69.6 | 50.7–82.6 | 50.2–51.2 | 82.1–83.1 |
| | Female | 798 | 68.5 | 46.3–81.4 | 45.8–46.8 | 80.9–81.9 |
| Overall | All gender | 1509 | 69.1 | 48.6–82.2 | 48.1–49.1 | 81.7–82.7 |
| **(B) (Monocytes % and granulocytes %)** | | | | | | |
| **Percent Monocytes (%)** | | | | | | |
| 1 to 6 months | Male | 362 | 7.6 | 3.9–13.29 | 3.4–4.4 | 12.79–13.79 |
| | Female | 394 | 7.4 | 3.38–13.52 | 2.85–3.85 | 13.02–14.02 |
| 6 to 12 months | Male | 200 | 7.2 | 3.0–12.8 | 2.5–3.5 | 12.3–13.3 |
| | Female | 187 | 6.7 | 3.3–11.8 | 2.8–3.8 | 11.3–12.3 |
| 12 to 17 months | Male | 149 | 7.0 | 3.0–15.7 | 2.5–3.5 | 15.2–16.2 |
| | Female | 217 | 6.8 | 2.8–12.2 | 2.3–3.3 | 11.7–12.7 |
| Overall | Male | 711 | 7.4 | 3.6–13.8 | 3.1–4.1 | 13.3–14.3 |
| | Female | 798 | 7.1 | 3.2–12.8 | 2.7–3.7 | 12.3–13.3 |
| Overall | All gender | 1509 | 7.2 | 3.4–13.3 | 2.9–3.9 | 12.8–13.8 |
| **Percent Granulocytes (%)** | | | | | | |
| 1 to 6 months | Male | 362 | 21.7 | 9.0–39.0 | 8.5–9.5 | 38.5–39.5 |
| | Female | 394 | 22.6 | 11.3–41.8 | 10.8–11.8 | 41.3–42.3 |
| 6 to 12 months | Male | 200 | 22.7 | 12.5–41.1 | 12.0–13.0 | 40.6–41.6 |
| | Female | 187 | 23.9 | 12.1–41.1 | 11.6–12.6 | 40.6–41.6 |
| 12 to 17 months | Male | 149 | 24.9 | 9.7–44.3 | 9.1–10.1 | 43.8–44.8 |
| | Female | 217 | 26.7 | 14.0–49.4 | 13.5–14.5 | 48.9–43.9 |
| Overall | Male | 711 | 22.6 | 9.7–41.8 | 9.2–10.2 | 41.3–42.3 |
| | Female | 798 | 23.8 | 11.6–44.7 | 11.1–12.1 | 44.2–45.2 |
| Overall | All gender | 1509 | 23.2 | 10.9–43.0 | 10.4–11.4 | 42.5–43.5 |
| **(C) (Red blood cells and hemoglobin)** | | | | | | |
| **Red Blood Cells(×10⁶ cells/µl)** | | | | | | |
| 1 to 6 months | Male | 362 | 4.02 | 2.73–5.37 | 2.23–3.23 | 4.87–5.87 |
| | Female | 394 | 4.02 | 2.87–5.40 | 2.37–3.37 | 4.90–5.90 |

*(Continued)*

**Table 2.** (Continued)

| Parameter/age group | | N | Median | 95% reference values | 90% Confidence Interval (Lower Limits) | 90% Confidence Interval (Higher Limits) |
|---|---|---|---|---|---|---|
| 6 to 12 months | Male | 200 | 4.89 | 3.43–5.91 | 2.93–3.93 | 5.41–6.41 |
| | Female | 187 | 4.68 | 3.65–5.86 | 3.15–4.15 | 5.36–6.36 |
| 12 to 17 months | Male | 149 | 4.83 | 3.29–6.08 | 2.79–3.79 | 5.58–6.58 |
| | Female | 217 | 4.80 | 3.23–5.86 | 2.73–3.73 | 5.36–6.36 |
| Overall | Male | 711 | 4.47 | 2.90–5.86 | 2.85–2.95 | 3.36–6.36 |
| | Female | 798 | 4.40 | 2.99–5.66 | 2.49–3.49 | 5.16–6.16 |
| Overall | All gender | 1509 | 4.44 | 2.94–5.78 | 2.44–3.44 | 5.28–6.28 |
| **Hemoglobin (g/dL)** | | | | | | |
| 1 to 6 months | Male | 362 | 11.4 | 8.5–15.0 | 8.0–9.0 | 14.5–15.5 |
| | Female | 394 | 11.7 | 8.6–15.4 | 8.1–9.1 | 14-9-15.9 |
| 6 to 12 months | Male | 200 | 10.3 | 7.5–12.4 | 7.0–8.0 | 13.9–14.9 |
| | Female | 187 | 10.3 | 7.9–13.0 | 7.4–8.4 | 12.5–13.5 |
| 12 to 17 months | Male | 149 | 10.0 | 6.6–12.6 | 6.1–7.1 | 12.1–13.1 |
| | Female | 217 | 10.2 | 6.9–12.6 | 6.4–7.4 | 12.1–13.1 |
| Overall | Male | 711 | 10.8 | 7.4–14.5 | 6.9–7.9 | 14.0–15.0 |
| | Female | 798 | 10.9 | 7.7–14.7 | 7.2–8.2 | 14.2–15.2 |
| Overall | All gender | 1509 | 10.9 | 7.6–14.6 | 7.1–8.1 | 14.1–15.1 |
| **(D) (Hematocrit and platelets)** | | | | | | |
| **Hematocrit (%)** | | | | | | |
| 1 to 6 months | Male | 362 | 34.5 | 26.4–45.8 | 25.9-26-9 | 45.3–46.3 |
| | Female | 394 | 35.4 | 26.6–46.9 | 26.1–27.1 | 46.4–47.4 |
| 6 to 12 months | Male | 200 | 32.2 | 24.0–38.2 | 24.0–25.0 | 37.7–38.7 |
| | Female | 187 | 32.3 | 25.5–38.8 | 25.0–26.0 | 38.3–39.3 |
| 12 to 17 months | Male | 149 | 31.4 | 21.7–38.7 | 21.2–22.2 | 38.2–39.2 |
| | Female | 217 | 32.1 | 23.3–38.9 | 22.8–23.8 | 38.4–39.4 |
| Overall | Male | 711 | 33.0 | 23.9–43.8 | 23.4–24.3 | 43.3–44.3 |
| | Female | 798 | 33.4 | 24.7–43.9 | 24.2–25.2 | 43.4–44.4 |
| Overall | All gender | 1509 | 33.2 | 24.4–43.8 | 23.9–24.9 | 43.3–44.3 |
| **Platelets (×10³ cells/µl)** | | | | | | |
| 1 to 6 months | Male | 362 | 341 | 136–704 | 135.5–136.5 | 703.5–704.5 |
| | Female | 394 | 382 | 160–695 | 159.5–160.5 | 694.5–695.5 |
| 6 to 12 months | Male | 200 | 469 | 119–771 | 118.5–119.5 | 119.0–120.0 |
| | Female | 187 | 456 | 98–816 | 97.5–98.5 | 98.0–99.0 |
| 12 to 17 months | Male | 149 | 415 | 93–808 | 92.5–93.5 | 807.5–808.5 |
| | Female | 217 | 418 | 112–776 | 111.5–112.5 | 775.5–776.5 |
| Overall | Male | 711 | 393 | 120–755 | 119.5–120.5 | 754.5–755.5 |
| | Female | 798 | 408 | 139–759 | 138.5–139.5 | 758.5–759.5 |
| Overall | All gender | 1509 | 400 | 129–757 | 128.5–129.5 | 756.5–757.5 |
| **(E) Erythrocyte indices (MCV and MCH)** | | | | | | |
| **Mean Cell Volume (fl)** | | | | | | |
| 1 to 6 months | Male | 362 | 91.8 | 60.6–110.4 | 60.1–61.6 | 109.9–110.9 |
| | Female | 394 | 92.3 | 63.7–109.7 | 63.2–64.2 | 109.2–110.2 |
| 6 to 12 months | Male | 200 | 66.2 | 53.6–77.5 | 53.1–54.1 | 77.0–78.0 |
| | Female | 187 | 68.6 | 52.34–82.16 | 51.84–52.84 | 81.7–82.7 |
| 12 to 17 months | Male | 149 | 64.6 | 52.8–79.93 | 52.3–53.8 | 79.4–80.4 |
| | Female | 217 | 67.9 | 53.76–80.76 | 53.26–54.26 | 80.3–81.3 |

(*Continued*)

**Table 2.** (Continued)

| Parameter/age group | | N | Median | 95% reference values | 90% Confidence Interval (Lower Limits) | 90% Confidence Interval (Higher Limits) |
|---|---|---|---|---|---|---|
| Overall | Male | 711 | 71.9 | 54.7–106.34 | 54.2–55.4 | 105.8–106.8 |
| | Female | 798 | 74.4 | 55.9–105.4 | 55.4–56.4 | 104.9–105.9 |
| Overall | All gender | 1509 | 73.4 | 54.9–105.6 | 54.4–55.4 | 105.1–106.1 |
| **Mean Cell Hemoglobin (pg)** | | | | | | |
| 1 to 6 months | Male | 362 | 30.1 | 19.6–36.7 | 19.1–20.1 | 36.2–37.2 |
| | Female | 394 | 30.8 | 20.4–36.2 | 19.9–20.9 | 35.7–36.7 |
| 6 to 12 months | Male | 200 | 21.3 | 16.22–25.9 | 15.72–16.72 | 25.4–26.9 |
| | Female | 187 | 22.3 | 16.64–27.62 | 16.14–17.14 | 27.12–28.12 |
| 12 to 17 months | Male | 149 | 20.5 | 15.38–26.25 | 14.88–15.88 | 25.75–26.75 |
| | Female | 217 | 21.7 | 16.15–26.76 | 15.65–16.55 | 26.26–27.26 |
| Overall | Male | 711 | 23.4 | 16.8–35.52 | 16.3–17.3 | 35.02–36.02 |
| | Female | 798 | 24.3 | 17.5–35.5 | 17.0–18.0 | 35.0–36.0 |
| Overall | All gender | 1509 | 24.0 | 16.9–35.5 | 16.4–17.4 | 35.0–36.0 |
| **(F) Lymphocyte and monocyte absolute counts** | | | | | | |
| **Lymphocyte absolute counts (×10³ cells/μl)** | | | | | | |
| 1 to 6 months | Male | 362 | 6.2 | 3.4–14.3 | 2.9–3.9 | 13.8–14.8 |
| | Female | 394 | 6.2 | 3.4–12.5 | 2.9–3.9 | 12.0–13.0 |
| 6 to 12 months | Male | 200 | 7.6 | 3.7–14.5 | 3.2–4.2 | 14.0–15.0 |
| | Female | 187 | 8.3 | 4.5–14.8 | 4.0–5.0 | 14.3–15.3 |
| 12 to 17 months | Male | 149 | 7.4 | 4.2–15.9 | 3.7–4.7 | 15.4–16.4 |
| | Female | 217 | 7.1 | 3.5–12.9 | 3.0–4.0 | 12.4–13.4 |
| Overall | Male | 711 | 6.7 | 3.6–14.5 | 3.1–4.1 | 14.0–15.0 |
| | Female | 798 | 6.8 | 3.4–13.2 | 2.9–3.9 | 12.7–13.7 |
| Overall | All gender | 1509 | 6.8 | 3.5–13.6 | 3.0–4.0 | 13.1–14.1 |
| **Monocyte absolute counts (×10³ cells/μl)** | | | | | | |
| 1 to 6 months | Male | 362 | 0.7 | 0.3–1.6 | 0.01–0.8 | 1.1–2.1 |
| | Female | 394 | 0.7 | 0.3–1.6 | 0.01–0.8 | 1.1–2.1 |
| 6 to 12 months | Male | 200 | 0.8 | 0.3–1.6 | 0.01–0.8 | 1.1–2.1 |
| | Female | 187 | 0.8 | 0.4–1.7 | 0.01–0.9 | 1.2–2.2 |
| 12 to 17 months | Male | 149 | 0.8 | 0.3–2.1 | 0.01–0.8 | 1.6–2.6 |
| | Female | 217 | 0.8 | 0.3–1.7 | 0.01–0.8 | 1.2–2.2 |
| Overall | Male | 711 | 0.8 | 0.3–1.6 | 0.01–0.8 | 1.1–2.1 |
| | Female | 798 | 0.7 | 0.3–1.6 | 0.01–0.8 | 1.1–2.1 |
| Overall | All gender | 1509 | 0.7 | 0.3–1.6 | 0.01–0.8 | 1.1–2.1 |
| **(G) Granulocyte absolute counts and MCHC** | | | | | | |
| **Granulocytes Absolute counts (×10³ cells/μl)** | | | | | | |
| 1 to 6 months | Male | 362 | 2.0 | 0.7–4.5 | 0.2–1.2 | 4.0–5.0 |
| | Female | 394 | 2.0 | 0.8–5.0 | 0.3–1.3 | 4.5–5.5 |
| 6 to 12 months | Male | 200 | 2.6 | 1.2–5.7 | 0.7–1.7 | 5.2–6.2 |
| | Female | 187 | 2.7 | 1.1–6.9 | 0.6–1.6 | 6.4–7.4 |
| 12 to 17 months | Male | 149 | 2.7 | 1.0–6.7 | 0.5–1.5 | 6.2–7.2 |
| | Female | 217 | 2.9 | 1.2–7.2 | 0.7–1.7 | 6.7–7.7 |
| Overall | Male | 711 | 2.3 | 0.8–5.6 | 0.3–1.3 | 5.1–6.1 |
| | Female | 798 | 2.4 | 0.9–6.3 | 0.4–1.4 | 5.8–6.8 |
| Overall | All gender | 1509 | 2.4 | 0.9–6.0 | 0.4–1.4 | 5.5–6.5 |
| **Mean Cell Hemoglobin Concentration (g/dl)** | | | | | | |

*(Continued)*

**Table 2.** (Continued)

| Parameter/age group | | N | Median | 95% reference values | 90% Confidence Interval (Lower Limits) | 90% Confidence Interval (Higher Limits) |
|---|---|---|---|---|---|---|
| 1 to 6 months | Male | 362 | 32.9 | 31.1–34.9 | 30.6–31.6 | 34.4–35.4 |
| | Female | 394 | 33.1 | 31.2–35.0 | 30.5–31.5 | 34.5–35.4 |
| 6 to 12 months | Male | 200 | 32.2 | 29.6–33.7 | 29.1–30.1 | 33.3–34.3 |
| | Female | 187 | 32.3 | 30.4–34.1 | 29.9–30.9 | 33.6–34.6 |
| 12 to 17 months | Male | 149 | 31.9 | 28.9–34.0 | 28.4–29.4 | 34.5–35.5 |
| | Female | 217 | 31.9 | 29.4–34.1 | 28.9–29.9 | 33.6–34.6 |
| Overall | Male | 711 | 32.5 | 29.8–34.7 | 29.3–30.3 | 34.2–35.2 |
| | Female | 798 | 32.6 | 30.0–34.7 | 29.5–30.5 | 34.2–35.2 |
| Overall | All gender | 1509 | 32.6 | 30.0–34.7 | 29.5–30.5 | 34.2–35.2 |
| **(H) MPV and RDW** | | | | | | |
| **Mean Platelet Volume** | | | | | | |
| 1 to 6 months | Male | 362 | 7.1 | 5.7–9.3 | 5.2–6.2 | 8.8–9.8 |
| | Female | 394 | 7.1 | 5.4–8.9 | 4.9–5.9 | 8.4–9.4 |
| 6 to 12 months | Male | 200 | 7.3 | 5.7–9.7 | 5.2–6.2 | 9.2–10.2 |
| | Female | 187 | 7.2 | 5.6–9.7 | 5.1–6.1 | 9.2–10.2 |
| 12 to 17 months | Male | 149 | 7.0 | 5.6–9.5 | 5.1–6.1 | 9.0–10.0 |
| | Female | 217 | 7.3 | 5.6–9.5 | 5.1–6.1 | 9.0–10.0 |
| Overall | Male | 711 | 7.1 | 5.7–9.5 | 5.2–6.2 | 9.0–10.0 |
| | Female | 798 | 7.1 | 5.6–9.4 | 5.1–6.1 | 8.9–9.9 |
| Overall | All gender | 1509 | 7.1 | 5.6–9.4 | 5.1–6.1 | 8.9–9.9 |
| **Red Cell Distribution Width** | | | | | | |
| 1 to 6 months | Male | 362 | 15.8 | 13.4–21.5 | 12.9–13.9 | 21.0–22.0 |
| | Female | 394 | 15.7 | 13.2–20.3 | 12.7–13.7 | 19.8–20.8 |
| 6 to 12 months | Male | 200 | 19 | 14.9–28.7 | 14.4–15.4 | 28.2–29.7 |
| | Female | 187 | 18.3 | 13.5–25.6 | 13.0–14.0 | 25.1–26.1 |
| 12 to 17 months | Male | 149 | 19.6 | 14.5–26.6 | 14.0–15.0 | 26.1–27.1 |
| | Female | 217 | 19.2 | 13.9–27.3 | 13.4–14.4 | 26.8–27.8 |
| Overall | Male | 711 | 17.3 | 13.7–25.3 | 13.2–14.2 | 24.8–25.8 |
| | Female | 798 | 16.7 | 13.3–25.1 | 12.8–13.8 | 24.6–25.6 |
| Overall | All gender | 1509 | 17.0 | 13.4–25.1 | 12.9–13.9 | 24.6–25.6 |

This table shows data for 95% reference values for white blood cells in different age categories and 90% confidence intervals stratified by gender.

The results in this table show 95% reference values for monocyte (%) and granulocyte (%) in different age categories and 90% confidence intervals stratified by gender.

The results in this table show 95% reference values for red blood cells and hemoglobin in different age categories and 90% confidence intervals stratified by gender.

The results in this table show 95% reference values for hematocrit and platelets in different age categories and 90% confidence intervals stratified by gender.

The results in this table show 95% reference values for MCV and MCH in different age categories and 90% confidence intervals stratified by gender.

The results in this table show 95% reference values for lymphocyte and monocyte absolute counts in different age categories and 90% confidence intervals stratified by gender.

The results in this table show 95% reference values for granulocyte absolute counts and MCHC in different age categories and 90% confidence intervals stratified by gender.

The results in this table show 95% reference values for MPV and RDW in different age categories and 90% confidence intervals stratified by gender.

For hemoglobin values, it was noted that in the 1m to 6m age groups, males had significantly lower median values than females (11.4g/dl versus 11.7g/dl in females, p = 0.0189). Similarly, when hemoglobin median values were compared for the overall study group, the median value in males and females were 10.8g/dl and 10.9g/dl (p = 0.0204) respectively.

For hematocrit levels in the overall group, the median value in females was 33.4% compared to males who had median value of 33.0%. (p = 0.0448).

For the 6m to 12m age group, there were differences in median RBC values males ($4.89\times10^6$ cells/μl) versus females ($4.68\times10^6$ cells/μl; p = 0.0065). Additionally, males had lower MCH values compared to females -21.3pg and 22.3pg, respectively (p<0.0001). This pattern was also observed in the 12m to 17m group -21.7pg for females versus 20.5pg for males (p = 0.0003) (Table 2E).

Median values for MCV were significantly higher in females (68.6fl) compared to males (66.2fl) in the 6m to 12m age group (p<0.0001). Similarly, for the overall study group, females had higher median MCV value (74.4fl) compared to males (71.9fl), (p = 0.0092).

## Hematology indices vary according to gender in children in Kisumu County

Granulocyte counts were higher in females compared to the males within all age groups. In the 12 to 17m age group, the granulocyte percent median in females was 26.7%, while that in males was 24.9% (p = 0.0456, Table 2B). Females were noted to have higher median platelets counts ($382\times10^3$ cells/μl) compared to that in males ($341\times10^3$ cells/μl; p<0.0005) in the 1m to 6m age group (Table 2D). In the combined age groups, the female granulocyte percent median was 23.8%, compared to the male granulocyte median which was 22.6%, p = 0.0044 (Table 3). While a few of the parameters showed differences, this was consistent with other studies that showed that for most parameters differences in regard to gender start being evident during adolescence [12].

Table 3 below shows a summary of the median values for the various analytes in the different age groups by gender.

**Median values of hematological indices vary across children of different age groups in Kisumu County.** *Median values and ranges for total White Blood Cell counts, Monocyte absolute counts and Granulocyte Absolute counts.* Children aged between 1–6 months had the lowest WBC counts ($9.1\times10^3$ cells/μl) which increased with each age group (p<0.0001, Table 4).

**Table 3. Summary of the median hematological and erythrocyte indices in the different age groups in relation to gender.**

| Analyte | Age Group 1m to 6m | | | Age Group 6m to 12m | | | Age Group 12m to 17m | | |
|---|---|---|---|---|---|---|---|---|---|
| | Male | Female | P value [a] | Male | Female | P value [a] | Male | Female | P value [a] |
| WBC | 8.9 | 9.2 | 0.6567 | 11.2 | 12 | 0.1129 | 11.2 | 10.9 | 0.5237 |
| LY% | 70 | 69.4 | 0.2370 | 69.8 | 70.1 | 0.8117 | 67.7 | 66.4 | 0.070 |
| MO% | 7.6 | 7.4 | 0.3423 | 7.2 | 6.7 | 0.0576 | 6.8 | 7.4 | 0.4721 |
| GRA% | 21.7 | 22.6 | 0.1202 | 22.7 | 23.9 | 0.6237 | 24.9 | 26.7 | **0.0456** |
| LY ABS | 6.2 | 6.2 | 0.9035 | 7.6 | 8.3 | 0.0957 | 7.4 | 7.1 | 0.3582 |
| MO ABS | 0.7 | 0.7 | 0.3865 | 0.8 | 0.8 | 0.6930 | 0.8 | 0.8 | 0.1756 |
| GRA ABS | 2.0 | 2.0 | 0.2915 | 2.6 | 2.7 | 0.1117 | 2.7 | 2.9 | 0.2400 |
| RBC | 4.02 | 4.02 | 0.3559 | 4.89 | 4.68 | **0.0065** | 4.83 | 4.80 | 0.1442 |
| HGB | 11.4 | 11.7 | **0.0189** | 10.3 | 10.3 | 0.2017 | 10.0 | 10.2 | 0.1387 |
| HCT | 34.5 | 35.4 | 0.0541 | 32.2 | 32.3 | 0.4460 | 31.4 | 32.1 | 0.1400 |
| MCV | 91.8 | 92.3 | 0.4701 | 66.2 | 68.6 | **<0.0001** | 64.6 | 67.9 | **<0.0001** |
| MCH | 30.1 | 30.8 | 0.2951 | 21.3 | 22.3 | **<0.0001** | 20.5 | 21.7 | **0.0003** |
| MCHC | 32.9 | 33.1 | 0.0554 | 32.2 | 32.3 | **0.0294** | 31.9 | 31.9 | |
| RDW | 15.8 | 15.7 | 0.0444 | 19.0 | 18.3 | 0.2842 | 19.6 | 19.2 | 0.1706 |
| MPV | 7.1 | 7.1 | 0.1987 | 7.3 | 7.2 | 0.1488 | 7.0 | 7.3 | 0.2402 |
| PLT | 341 | 382 | **<0.0005** | 469 | 456 | 0.2739 | 415 | 418 | 0.8831 |

This table shows a summary of the median values in different hematological and erythrocyte indices in different age groups by gender.

[a] Comparisons between males and females. **P values in bold represent statistically significant results**.

**Table 4. Summary of the median values for the various hematological analytes in the different age groups.**

| Analyte | Age groups | | | |
|---|---|---|---|---|
| | 1m to 6m (G1) | 6m to 12m (G2) | 12m to 17m (G3) | (Groups with P value <0.001) |
| WBC | 9.1 | 11.6 | 11.1 | (G1 and G2; G1 and G3) |
| LY% | 69.8 | 69.9 | 66.8 | (G1 and G3; G2 and G3) |
| MO% | 7.4 | 6.9 | 6.9 | (G1 and G2; G1 and G3) |
| GRA% | 22.1 | 23.1 | 26.3 | (G1 and G3; G2 and G3) |
| LY ABS | 6.2 | 8 | 7.2 | (All groups) |
| MO ABS | 0.7 | 0.8 | 0.8 | (G1 and G2; G1 and G3) |
| GRA ABS | 2 | 2.6 | 2.85 | (G1 and G2; G1 and G3) |
| RBC | 4.02 | 4.83 | 4.81 | (G1 and G2; G1 and G3) |
| HGB | 11.6 | 10.3 | 10.2 | (G1 and G2; G1 and G3) |
| HCT | 35.1 | 32.2 | 31.8 | (G1 and G2; G1 and G3) |
| MCV | 92.1 | 67.4 | 66.1 | (G1 and G2; G1 and G3) |
| MCH | 30.5 | 21.7 | 21.1 | (G1 and G2; G1 and G3) |
| MCHC | 33 | 32.3 | 31.9 | (G1 and G2; G1 and G3) |
| RDW | 15.7 | 18.8 | 19.5 | (G1 and G2; G1 and G3) |
| PLT | 357 | 465 | 416 | (All groups) |

Summary of results showing median values of hematological and erythrocyte indices across different age groups.

Absolute monocyte counts in the 1m to 6m age group was 0.7 (x$10^3$ cells/μl, both the median value in 6m to 12m and 12m to 17m was 0.8 (×$10^3$ cells/μl). For granulocyte absolute counts, the median value in 1m to 6m age group was 2.0 (×$10^3$ cells/μl), the median value in 6m to 12m age group was 2.6 while the median value in 12m to 17m age group was 2.85 (p<0.0001, Table 4).

**Median values and ranges for granulocyte percent and lymphocyte percent.** Median granulocyte percent increased with increasing age (*p<0.0001*). Children aged 1-6m had the lowest percent granulocyte (22.1%) followed by 6-12months age group (23.1%) and lastly 12m to 17m (26.3%) (Table 4). Median lymphocytes percent were noted to be lower in the upper age group (12–17 months) compared to the two lower age categories. Children aged 1 to 6m had median lymphocyte percent of 69.8%, those aged 6-12months had median lymphocyte counts of 69.9% while those aged 12m to 17m had lymphocyte counts of 66.8% (p<0.0001) (Table 4).

**Median values and ranges of absolute lymphocytes counts in children of different age categories.** Median absolute lymphocyte counts were noted to be significantly different from all the three age categories with lower values (6.2×$10^3$ cells/μl) in the lowest age category (1m to 6m) and higher values in the 6-12m age category (8×$10^3$ cells/μl) (p<0.0001). Children aged 12m to 17m had median value of 7.2x$10^3$ cells/μl.

In 1m to 6m age group, the median value of the percent monocytes was statistically significantly higher compared to the age groups 6m to 12m and 12m to 17m *(p<0.0001)*. Median value in 1m to 6m age group was 7.4%, children aged 6m to 12m and 12m to 17m each had median values of 6.9%.

**Median values of erythrocyte indices vary across children of different age groups in Kisumu County.** It was noted in 1m to 6m age group, the median values of the hemoglobin and red cell indices are statistically significantly higher compared to the age groups 6m to 12m and 12m to 17m (p<0.0001). The median values for hemoglobin in children aged 1m to 6m was 11.6g/dl compared to median values in children aged 6m to 12m and 12m to 17m who had median values of 10.3 g/dl and 10.2 g/dl respectively. Children aged 1m to 6m had

hematocrit median values of 35.1% compared to children aged 6m to 12m and 12m to 17m who had median values of 32.2% and 31.8% respectively (Table 4). Children aged 1m to 6m had MCV median values of 92.1fl compared to children aged 6m to 12m and 12m to 17m who had median values of 67.4fl and 66.1fl respectively. For MCHC, the values were noted to be decreasing as per increasing age and this was different in all the three age groups (p<0.0001). Children aged 1m to 6m had MCHC median values of 33g/dl, while children aged 6m to 12m had MCHC median values of 32.3 g/dl. Children aged 12m to 17m had the least MCHC median values of 31.9g/dl (Table 4).

Red blood cells and RDW in 1m to 6m age groups were statistically significantly different lower than other two age groups (*p<0.001*). No differences were observed in the other age categories 6-12m and 12-17m for these analytes. Children aged 1m to 6m had RBC median values of $4.02 \times 10^6$ cells/μl, followed by children aged 12m to 17m who had median values of $4.81 \times 10^6$ cells/μl and lastly children aged 6m to 12m who had median values of $4.83 \times 10^6$ cells/μl (Table 4).

It was noted that for platelets counts, the values were significantly different in all the three age categories with lower values in the lowest age category (1m to 6m) and higher values in the 6-12m age category (*p<0.001*). Children aged 1m to 6m had median platelets counts of $357 \times 10^3$ cells/μl followed by children aged 12m to 17m who had median platelet counts of $416 \times 10^3$ cells/μl and lastly children aged 6m to 12m had median platelet counts of $465 \times 10^3$ cells/μl (Table 4).

Table 4 shows the summary of the median values for the various analytes in the different age groups.

## Current reference ranges for this study compared to other available reference values

Tables 5, 6 and 7 show the 95% reference ranges of the current study in Kisumu for the various age groups compared to the other available reference values. Despite Kilifi and Kisumu being in the same country, there were noted some differences in our 2.5[th] and 97.5[th] percentiles compared to the percentiles in the children of the same age category in Kilifi. Table 5 shows 95% reference intervals for hematological parameters for Kisumu infants aged 1m to 6months, compared to published data from Kilifi for infants aged 1m to 6months. In the 1m to 6m age group, both 2.5[th] and 97.5[th] percentiles for WBCs, hemoglobin, hematocrit, MCV and lymphocytes absolute count values for Kisumu children were noted to be higher than the Kilifi values. Both the 2.5[th] and 97.5[th] percentile values for absolute monocyte counts in Kisumu infants were lower compared to the Kilifi infants. Ranges for platelets were narrower in Kisumu compared to Kilifi values. For the overall cohort, the same pattern as for the 1m to 6m age group was depicted whereby Kisumu 2.5[th] and 97.5[th] percentiles for WBCs, hemoglobin, hematocrit, MCV and lymphocytes absolute count values were higher compared to Kilifi values. Similarly, the platelet ranges were narrower for Kisumu infants compared to Kilifi values.

Table 6 shows the 95% reference intervals for hematological parameters for Kisumu infants aged 1m to less than 12 months and 1m to 17 months, compared to published data from Kilifi. Table 7 shows the 95% reference intervals for hematological parameters for Kisumu infants aged 1m to less than 12months, compared to published data from Kilifi, Tanzania and United States/Europe.

## Discussion

Reference values have a significant bearing on interpretation of test results [13]. The health of an individual can be readily assessed by data provided from the laboratory investigations.

**Table 5. Comparison of 95% hematological reference intervals for infants aged 1m to 6months between Kisumu and Kilifi.**

| Parameter | 95% reference intervals in males | | | 95% reference intervals in females | |
|---|---|---|---|---|---|
| | Kisumu (current) | Kilifi | | Kisumu (current) | Kilifi |
| White Blood Cells (×10³ cells/µl) | 5.2–19.5 | 4.60–13.71 | | 5.0–16.9 | 5.01–15.93 |
| Lymphocytes (%) | 52.5–83.6 | - | | 49.1–81.8 | - |
| Monocytes (%) | 3.9–13.3 | - | | 3.4–13.5 | - |
| Granulocytes (%) | 9.0–39.0 | - | | 11.3–41.8 | - |
| Red Blood Cells (×10⁶ cells/µl) | 2.73–5.37 | - | | 2.87–5.40 | - |
| HGB (g/dl) | 8.5–15.0 | 8.0–14.0 | | 8.6–15.4 | 8.3–13.8 |
| HCT (%) | 26.4–45.8 | 24.6–41.9 | | 26.6–46.9 | 25.2–42.5 |
| Platelets (×10³ cells/µl) | 136–704 | 93–746 | | 160–695 | 22–833 |
| MCV (fl) | 60.6–110.4 | 58–98 | | 63.7–109.7 | 55–102 |
| MCH (pg) | 19.6–36.7 | - | | 20.4–36.2 | - |
| MCHC (g/dl) | 31.1–34.9 | 31.2–34.7 | | 31.2–35.0 | 31.0–35.1 |
| MPV | 5.7–9.3 | - | | 5.4–8.9 | - |
| RDW | 13.4–21.5 | - | | 13.2–20.3 | - |
| LY# (×10³ cells/µl) | 3.4–14.4 | 2.25–8.99 | | 3.4–12.5 | 3.39–9.20 |
| MO# (×10³ cells/µl) | 0.3–1.6 | 0.37–1.88 | | 0.3–1.6 | 0.35–1.91 |
| GRA# (×10³ cells/µL) | 0.7–4.5 | - | | 0.9–6.0 | - |

Table showing comparison of Kisumu ranges against Kilifi ranges for 1m to 6m age group.

Some data were missing from published data for Kilifi values to compare with Kisumu reference ranges.

**Table 6. Comparison of 95% hematological reference intervals for infants aged 1m to less than 12 months and 1m to 17months between Kisumu and Kilifi.**

| Parameter | 95% reference intervals in children aged 1m to less than 12months | | | 95% reference intervals in children aged 1m to 17months | |
|---|---|---|---|---|---|
| | Kisumu (current) | Kilifi | | Kisumu (current) | Kilifi |
| White Blood Cells (×10³ cells/µl) | 9.7 (5.4–19.2 | 9.7 (5.6–16.6) | | 10.1 (5.7–19.8) | 10.0 (5.71–16.72) |
| Lymphocytes (%) | 69.9 (51.1–82.6) | - | | 69.1 (48.6–82.2) | - |
| Monocytes (%) | 7.3 (3.7–13.2) | - | | 7.2 (3.4–13.3) | - |
| Granulocytes (%) | 22.3 (10.3–40.7) | - | | 23.2 (10.9–43.0) | - |
| Red Blood Cells (×10⁶ cells/µl) | 4.26 (2.90–5.58) | - | | 4.44 (2.94–5.78) | - |
| HGB (g/dl) | 11.1 (8.1–14.9) | 9.8 (7.3–13.2) | | 10.9 (7.6–14.6) | 9.8 (7.2–12.7) |
| HCT (%) | 34 (25.5–44.7) | 30.6 (23.5–39.2) | | 33.2 (24.4–43.8) | 30.6 (23.9–38.3) |
| Platelets (×10³ cells/µl) | 393 (147–734) | 451 (72.7–769.2) | | 400 (129–757) | 462 (84–773) |
| MCV (fl) | 86.3 (57.1–108.1) | 72 (53.4–98.6) | | 73.4 (54.9–105.6) | 69 (52–97) |
| MCH (pg) | 28.1 (18.1–36.1) | - | | 24 (16.9–35.5) | - |
| MCHC (g/dl) | 32.8 (30.7–34.8) | - | | 32.6 (30.0–34.7) | 32.1 (29.4–34.4) |
| MPV | 7.1 (5.6–9.4) | - | | 7.1 (5.6–9.4) | - |
| RDW | 16.2 (13.3–23.0) | - | | 17 (13.4–25.1) | - |
| LY# (×10³ cells/µl) | 6.6 (3.4–13.5) | 5.96 (3.3–10.2) | | 3.5–13.6 | 6.0 (3.13–10.2) |
| MO# (×10³ cells/µl) | 0.7 (0.3–1.6) | 1.02 (0.5–2.0) | | 0.3–1.6 | 1.02 (0.48–1.93) |
| GRA# (×10³ cells/µl) | 2.2 (0.8–5.3) | - | | 0.9–6.0 | - |

Table showing comparison of Kisumu and Kilifi hematological median values.

Some parameters were not available in Kilifi data for comparison with Kisumu results.

**Table 7. Comparison of 95% hematological reference intervals for infants aged 1m to less than 12months between Kisumu, Kilifi, Tanzania and United States/Europe.**

| Parameter | Kisumu (current) | Kilifi | Tanzania | USA/Europe |
|---|---|---|---|---|
| WBC ($\times 10^3$ cells/μl) | 9.7 (5.4–19.2) | 5.6–16.6 | 2.0–17.3 | 5.0–17.0 |
| Lymphocytes (%) | 69.9 (51.1–82.6) | - | - | - |
| Monocytes (%) | 7.3 (3.7–13.2) | - | - | - |
| Granulocytes (%) | 22.3 (10.3–40.7) | - | - | - |
| RBC ($\times 10^6$ cells/μl) | 4.26 (2.91–5.58) | - | - | - |
| HGB (g/dl) | 11.1 (8.1–14.9) | 7.3–13.2 | 8.1–13.2 | 9.4–13.0 |
| HCT (%) | 34 (25.5–44.7) | 23.5–39.2 | 25.1–38.6 | 28–42 |
| Platelets ($\times 10^3$ cells/μl) | 393 (147–734) | 72.7–769.2 | 25–708 | 150–400 |
| MCV (fl) | 86.3 (57.1–108.1) | 53.4–98.6 | 53.3–96.6 | 70–98 |
| MCH (pg) | 28.1 (18.1–38.1) | - | - | - |
| MCHC (g/dl) | 32.8 (30.7–34.8) | - | - | - |
| MPV | 7.1 (5.6–9.4) | - | - | - |
| RDW | 16.2 (13.3–23.0) | - | - | - |
| LY# ($\times 10^3$ cells/μl) | 6.6 (3.4–13.5) | 3.3–10.2 | 3.3–11.8 | 3.3–11.5 |
| MO# ($\times 10^3$ cells/μl) | 0.7 (0.3–1.6) | 0.5–2.0 | 0.2–1.5 | 0.2–1.3 |
| GRA# ($\times 10^3$ cells/μl) | 2.2 (0.8–5.3) | - | - | - |

Table showing comparison of Kisumu hematological reference ranges with some published data.

Some data were missing from other published data to compare with Kisumu reference ranges.

Hematological evaluations provide a quick preview of a patient's well-being and are very important in making clinical decisions [1–5]. These values are frequently used in clinical management, clinical trials and research studies for monitoring patient and participant's health [1, 8]. If national, regional or local reference values are unavailable, clinical care providers usually revert to those established in America and Europe, which are often different to the applied population. There could be many variations in such values due to age, nutrition, genetic differences, exposure to infectious diseases, ethnic origins, socio-demographic characteristics and other factors based on the environment [8, 9]. It is thus beneficial to develop regional and age specific reference values that can be applied for efficient patient management and improved conduct of clinical research.

This study was carried out in Kombewa Sub-County in Kisumu County, Western Kenya, a region that has extensively benefited from many clinical research studies. The aim of our study was to establish locally derived hematological reference values for infants aged 1 month to 17 months of age, which can serve as standards for interpretation of clinical hematology laboratory results. The values currently mostly used in Kombewa Sub-County for the infants are those from the Harriet Lane Handbook [14] and the manufacturer of the hematology analyzer used. This study allowed us to develop reference ranges more relevant to pediatric medicine in Kenya. The results describe participants evaluated as "normal" through predetermined criteria including physical examination.

Despite Kilifi and Kisumu both being in the same country in malaria endemic regions [15, 16], there were noted differences between our 95% reference intervals compared to those for the same age category in Kilifi. Generally, it was observed that Kisumu infants had higher lower and higher percentiles compared to those in Kilifi. These differences may be due to differences in altitude, differences in regard to exposure to infectious diseases, ethnic constitution of the study population, differences in nutrition or other environmental factors.

Kisumu residents are mainly Luo whose mainstay economic activity is fishing while the Kilifi community is mainly Swahili and agriculturalists. Kisumu region is located on the equator in a rich agricultural area and fish is the staple food, which places children in Kisumu at a better nutritional status than Kilifi. Compared to Tanzanian hematology reference ranges, Kisumu hematology reference limits were observed to be higher than for WBC, absolute lymphocyte and monocyte counts, hemoglobin, hematocrit and MCV. In our study, we observed higher ranges of WBCs, absolute lymphocyte and monocyte counts compared to values in US and Europe. This would imply cases that would be considered as leukocytosis if American and/or European values were used, would actually be normal using the Kisumu reference range. Several African studies have shown lower red blood cell indices in children and adults. In this study we observed wider ranges in hemoglobin, hematocrit, and MCV. This could be attributed to factors such as malaria, other parasitic infections, haemoglobinopathies and iron deficiency anemia [15]. Thus, if American or European hematological ranges were used in Kisumu County rather than locally derived, participants who would otherwise have been enrolled into clinical trials/studies as healthy would be classified as unhealthy and may have been excluded from trial participation. Wider ranges were observed in platelet counts in Kisumu children compared to the US/Europe ranges. Thus, using US/European values in determining eligibility for study enrolment in Kisumu, healthy participants would have been excluded on the basis of being considered as having thrombocytopenia or thrombocytosis, confirming the necessity to apply regional or local reference ranges.

The analyzer used for this study could not determine basophils, eosinophils, and neutrophils separately rather grouping these as granulocytes. There were no values for RBC and other RBC indices from Kilifi, Tanzania and America/European populations that could be used to compare the Kombewa values.

## Strengths and limitations

One major strength of our study is that we analyzed a large sample size (n = 1509) of infants in a community setting. We assert that these reference ranges are representative of infants in similar settings within Africa who would ultimately benefit from successful drug or vaccine studies. Data analysis was robust and comparison of the values with other published data was vigorously done.

Some limitations were noted in the current study. In as much as proper medical examinations were carried out for the children from whom these data were drawn, some sub-clinical conditions were not established that included screening for some diseases e.g. hepatitis B, hepatitis C and syphilis. However, the infants were tested on case by case through maternal exposure and those who were exposed were excluded. The analysis, moreover, excluded data for the participants with signs or symptoms of acute illness and so in the context of resource-limited settings, we consider the data representative of healthy infants from Kombewa Sub-County.

## Conclusion

The current study provides the first locally established clinical hematological laboratory reference ranges in infants from 1m to 17 m from Kombewa Sub-County, Western Kenya. The established values are useful in providing an important tool for patient management. Additionally, they influence decisions on inclusion of participants as well as improving scientific validity in clinical trials conducted locally in children less than two years of age. These locally derived hematology reference ranges would be more appropriate for use in the Kisumu population than those adopted from other countries. The findings above underscore the importance

of using age specific reference ranges in children and may be most applicable to be currently adopted in Kombewa Sub-County.

## Supporting information

**S1 Data. Minimal data set.**
(XLS)

**S1 File. Mal-055 protocol.**
(PDF)

**S2 File. Approval letter for use of data.**
(PDF)

**S3 File. Waiver of ICF.**
(DOC)

**S4 File. Protocol 2325 WRAIR IRB approval and commander implementation.**
(PDF)

**S5 File. WRAIR IRB approval.**
(PDF)

## Acknowledgments

The authors would like to thank the study participants who participated in the primary phase III RTS,S vaccine study thus allowing data generation. We also thank the sponsor, GSK for granting permission to use the data.

## Author Contributions

**Conceptualization:** Jew Ochola Ouma, David H. Mulama, Lucas Otieno, Bernhards Ogutu, Janet Oyieko, Walter Otieno.

**Formal analysis:** Jew Ochola Ouma, Janet Oyieko.

**Investigation:** Jew Ochola Ouma, Lucas Otieno, John Owuoth, Bernhards Ogutu, Janet Oyieko, Ben Andagalu.

**Methodology:** Jew Ochola Ouma, Janet Oyieko, Peter Sifuna, Victorine Owira.

**Project administration:** Valentine Singoei, Victorine Owira, Stacey Maureen Okallo Gondii.

**Resources:** Walter Otieno.

**Supervision:** David H. Mulama, John Owuoth, Bernhards Ogutu, Janet Oyieko, Jackson C. Korir, Walter Otieno.

**Writing – original draft:** Jew Ochola Ouma, David H. Mulama, Lucas Otieno, Jackson C. Korir, Ben Andagalu, Walter Otieno.

**Writing – review & editing:** Jew Ochola Ouma, David H. Mulama, Janet Oyieko, Jackson C. Korir.

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
