## [Decision Letter · Decision Letter 0]

13 Mar 2020

PONE-D-19-28401

Clinical Laboratory Hematology Reference Values among children aged 4 weeks to 17months in Kombewa Sub-County, Kisumu: A cross sectional study of rural population in Western Kenya

PLOS ONE

Dear Mr Ochola,

Thank you for submitting your manuscript to PLOS ONE. After careful consideration, we feel that it has merit but does not fully meet PLOS ONE’s publication criteria as it currently stands. Therefore, we invite you to submit a revised version of the manuscript that addresses the points raised during the review process.

The manuscript has been evaluated by two reviewers, including a statistical reviewer, and their comments are available below.

The reviewers have raised a number of concerns. They feel the statistical analyses performed in this manuscript should be revisited and explained more clearly and potentially re-analysed due to the lack of normally distributed data. In addition, the conclusion and limitations sections need to be discussed in further detail and there are some errors of language that remain.

Could you please carefully revise the manuscript to address all comments raised?

We would appreciate receiving your revised manuscript by Apr 25 2020 11:59PM. To enhance the reproducibility of your results, we recommend that if applicable you deposit your laboratory protocols in protocols.io, where a protocol can be assigned its own identifier (DOI) such that it can be cited independently in the future. For instructions see: http://journals.plos.org/plosone/s/submission-guidelines#loc-laboratory-protocols

We look forward to receiving your revised manuscript.

Kind regards,

Natasha Rickett

Academic Editor

PLOS ONE

Journal Requirements:

"The funders had no role in study design, data collection and analysis, decision to publish, or preparation of the manuscript"

a)    Please provide an amended Funding Statement that declares *all* the funding or sources of support received during this specific study (whether external or internal to your organization) as detailed online in our guide for authors at http://journals.plos.org/plosone/s/submit-now.  

b)    Please state what role the funders took in the study.  If any authors received a salary from any of your funders, please state which authors and which funder. If the funders had no role, please state: "The funders had no role in study design, data collection and analysis, decision to publish, or preparation of the manuscript."

4. We note you have included a table to which you do not refer in the text of your manuscript. Please ensure that you refer to Table 6-7 in your text; if accepted, production will need this reference to link the reader to the Table.

Reviewers' comments:

Reviewer's Responses to Questions

**Comments to the Author**

1. Is the manuscript technically sound, and do the data support the conclusions?

Reviewer #1: Partly

Reviewer #2: Yes

2. Has the statistical analysis been performed appropriately and rigorously? 

Reviewer #1: No

Reviewer #2: Yes

3. Have the authors made all data underlying the findings in their manuscript fully available?

Reviewer #1: Yes

Reviewer #2: Yes

4. Is the manuscript presented in an intelligible fashion and written in standard English?

Reviewer #1: No

Reviewer #2: Yes

5. Review Comments to the Author

Reviewer #1: This paper is a resubmission of a previous paper on haematological reference ranges in Kenya.

The authors I see have been asked to improve the English - however, I note that the first sentence of the abstract is incorrect. Populations here must be plural, and local isn't contextualised here. The issue continues - textbooks cannot do things as they are inanimate, population is not pluralised when required etc etc. A careful proof read here is required and editing to ensure proper English.

The study itself is a nested cross sectional study within a drug-company trial of a malaria vaccine. There are two groups in this trial, one aged 6-12 weeks and one aged 5-17 months. It does make sense for analyses to be stratified by the two different groups here because the groups according to clinicaltrials.gov would represent two similar but distinct randomisations and the methodologies of large collaborations such as the EBCTCG should be used to stratify for group. The gap in enrollemnt ages here would tend to imply that the age groups used here may not be appropriate and a histogram of ages should be given to show what happens between 12 and 21 weeks of age.

The trial data posted on the clinicaltrials.gove dataset gives very different numbers - pleas explain how the numbers relate to the 15,459 recruited - what proportion of participants in this province contribute data here?

Given the confidence interval around the reference ranges, what is the suggested normal range - is it least lower limt to most higher limit?

Star notation has been outmoded since 1990 or thereabouts - please do not use it in the Tables, and explain what is being tested? Is it male vs female?

Please regress age to look at its effect as a continuous variable - the Mann-Whitney U test or its extension to Keruskal-Wallis is not appropriate for a continuous independent variable. Analyses need to be stratified for trial stratum.

The Mann-Whitney U test does not compare medians - it compares distributions.

What adjustment is made for multiple testing here?

Please explain how data for Kilifi was made available at an IPD level to do the tests as the legend talks of published reference data. Surely the issue is the size of any difference and in particular the difference in variance and whether there is any clinical relevance to any differences as opposed to a simple statistical test.

Reviewer #2: The article is well written and is of great relevance for health teams in the management of patients in the region.

I observed some redundancies such as the statistical explanation and the objectives that were repeated in the method, result and also in the discussion. However, it does not remove the value of the work.

1- The result of the summary left something to be desired, since the reader already wants in the summary to know a preview of the observed reference intervals. There was an emphasis on showing the differences. However, this is not the main objective of the work.

It is important to emphasize the peculiarities of the reference values in this population. In summary, for this work to be showing the value of "p" on the differences is unnecessary.

In the summary, focus on answering the objective.

2- In methods the part that says, "Despite various malaria interventions by Kenya Medical research institute, malaria transmission and intensity still remains high", should go into limitations.

3- The author states that they are healthy patients, "A qualified physician attached to the project performed physical assessment of all the participants and only healthy participants were recruited", however it is necessary to mention which criteria this doctor used to consider a healthy patient.

4- Make it clear if the sample is random or if it was obtained for convenience. As part of a database already available, how was the recruitment of these patients? Was it random or not? Explain.

4 - The statistical analysis is robust and is based on non-parametric data. Explain which analyzes were performed to consider the data as non-parametric. Ideally, they should be parametric as there are many participants. Not being parametric, it shows that the values are not well distributed around an average. Shows that the data does not have a normal curve. In this case, put that in the LIMITATIONS.

5- I suggest showing a histogram with the age distribution by age group.

6- In results explain the reason for the exclusion of 122 patients.

7- Present a table with the reference intervals that can be easily used by academics and health professionals. In this table there is no need for a "p" value. Display only the parameter and its reference range. This table will be the author's main contribution to the scientific community.

6. PLOS authors have the option to publish the peer review history of their article (what does this mean?). If published, this will include your full peer review and any attached files.

Reviewer #1: No

Reviewer #2: Yes: Eduardo Rodrigues Alves Junior

---

## [Author Response · Author response to Decision Letter 0]

27 Jul 2020

REBUTTAL LETTER 6- 17July2020 

• A rebuttal letter that responds to each point raised by the academic editor and reviewer(s). This letter should be uploaded as separate file and labeled 'Response to Reviewers'.

• A marked-up copy of your manuscript that highlights changes made to the original version. This file should be uploaded as separate file and labeled 'Revised Manuscript with Track Changes'.

• An unmarked version of your revised paper without tracked changes. This file should be uploaded as separate file and labeled 'Manuscript'.

 Response- My ORCID iD is 0000-0002-6356-6875 

"The funders had no role in study design, data collection and analysis, decision to publish, or preparation of the manuscript"

a) Please provide an amended Funding Statement that declares *all* the funding or sources of support received during this specific study (whether external or internal to your organization) as detailed online in our guide for authors at http://journals.plos.org/plosone/s/submit-now. 

 b) Please state what role the funders took in the study. If any authors received a salary from any of your funders, please state which authors and which funder. If the funders had no role, please state: "The funders had no role in study design, data collection and analysis, decision to publish, or preparation of the manuscript."

 Response- We thank the reviewer for this comment -this has been added in the cover letter. Quoted as follows;

 ‘’The funders had no role in study design, data collection and analysis, decision to publish, or preparation of the manuscript.’’ The Principal Investigator of the study as well did not receive any salary for this funding

4. We note you have included a table to which you do not refer in the text of your manuscript. Please ensure that you refer to Table 6-7 in your text; if accepted, production will need this reference to link the reader to the Table.

 Response- Thank you for pointing out table 6 and 7 which shows the 95% reference intervals for hematological parameters for Kisumu children compared to published data from Kilifi, and the 95% reference intervals for hematological parameters for Kisumu children compared to published data from Kilifi, Tanzania as well as United States/Europe respectively. We have revised and referred the table into the text in both in the tracked and clean manuscripts. It is found in line 517 of the tracked manuscript and 350 of the clean manuscript.

Reviewers' comments:

Reviewer's Responses to Questions

Comments to the Author

1. Is the manuscript technically sound, and do the data support the conclusions?

Reviewer #1: Partly We appreciate your insights. We have incorporated all the suggestions to make this manuscript technically sound 

Reviewer #2: Yes We thank you for the complement ________________________________________ 2. Has the statistical analysis been performed appropriately and rigorously? 

Reviewer #1: No. We thank you for the comment. We have again re-verified on statistical analysis as were done and have incorporated all the necessary explanations and suggestions 

Reviewer #2: Yes We thank you for the complement 

 3. Have the authors made all data underlying the findings in their manuscript fully available?

Reviewer #1: Yes

Reviewer #2: Yes 

 Response-we thank you for seeing this. The data has been re-attached as one of the supporting documents in the submission in excel format. It is named ‘hematology reference range data’ ________________________________________ 4. Is the manuscript presented in an intelligible fashion and written in standard English?

Reviewer #1: No Response-we thank you for the comment. We have proofread the paper and even enlisted services of a scientific writer to check grammatical errors in the manuscript.

 Reviewer #2: Yes- We thank you for the complement ________________________________________ 5. Review Comments to the Author

Reviewer #1: This paper is a resubmission of a previous paper on haematological reference ranges in Kenya.

The authors I see have been asked to improve the English - however, I note that the first sentence of the abstract is incorrect. Populations here must be plural, and local isn't contextualised here. The issue continues - textbooks cannot do things as they are inanimate, population is not pluralised when required etc etc. A careful proof read here is required and editing to ensure proper English.

The study itself is a nested cross-sectional study within a drug-company trial of a malaria vaccine. There are two groups in this trial, one aged 6-12 weeks and one aged 5-17 months. It does make sense for analyses to be stratified by the two different groups here because the groups according to clinicaltrials.gov would represent two similar but distinct randomizations and the methodologies of large collaborations such as the EBCTCG should be used to stratify for group. The gap in enrollment ages here would tend to imply that the age groups used here may not be appropriate and a histogram of ages should be given to show what happens between 12 and 21 weeks of age. 

The trial data posted on the clinical trials.gov dataset gives very different numbers - please explain how the numbers relate to the 15,459 recruited - what proportion of participants in this province contribute data here? 

 Response- We thank you for the comment. The dataset in the clinical trials.gov is the total of the number of children that were recruited in the vaccine trial arms in the different countries in different sites. This was a total children in 11 sites in the 7 countries that participated in the malaria vaccine trial. The analysis for the current manuscript is limited only to the study participants enrolled in the Kombewa study as per the objective for the reference ranges for that particular sub-county. In the study, only 1509 children who met the study inclusion criteria were enrolled and thus the sample size that was used for data analysis. 

Given the confidence interval around the reference ranges, what is the suggested normal range - is it least lower limit to most higher limit?

 Response-Normal range value will coalesce around the median at 95% CI and will be 5% lower or higher as indicated in our table 2 

Star notation has been outmoded since 1990 or thereabouts - please do not use it in the Tables, and explain what is being tested? Is it male vs female?

 Response- The star notation has been removed and where values were significant, the values have bolded and appropriately explained in the text. 

Please regress age to look at its effect as a continuous variable - the Mann-Whitney U test or its extension to Keruskal-Wallis is not appropriate for a continuous independent variable. Analyses need to be stratified for trial stratum. The Mann-Whitney U test does not compare medians - it compares distributions. What adjustment is made for multiple testing here?

 Response- We thank the reviewer for pointing out the uses for mann whitney u test. To our understanding, we used the test to compare the variables for two groups. This test is used for both continuous and non-continuous variables. 

Please explain how data for Kilifi was made available at an IPD level to do the tests as the legend talks of published reference data. Surely the issue is the size of any difference and in particular the difference in variance and whether there is any clinical relevance to any differences as opposed to a simple statistical test. 

 Response- We thank you for this comment. This was in relation to comparison of both the lower (2.5th) and upper (97.5th) percentiles for the variables analytes described. Kilifi has the published ranges with these limits. These statements regarding the comparison that was done has been made clearer in regard to the upper and lower percentile limits and ranges. To the best of our knowledge, such variance especially if both the lower and upper limits are lower or higher than the comparator, this is of clinical significance and should be mentioned. 

Reviewer #2: The article is well written and is of great relevance for health teams in the management of patients in the region.

I observed some redundancies such as the statistical explanation and the objectives that were repeated in the method, result and also in the discussion. However, it does not remove the value of the work.

1- The result of the summary left something to be desired, since the reader already wants in the summary to know a preview of the observed reference intervals. There was an emphasis on showing the differences. However, this is not the main objective of the work.

It is important to emphasize the peculiarities of the reference values in this population. In summary, for this work to be showing the value of "p" on the differences is unnecessary.

In the summary, focus on answering the objective.

 Response- We thank you for this comment. This has been rephrased and the P-values in the reference ranges have been removed 

2- In methods the part that says, "Despite various malaria interventions by Kenya Medical research institute, malaria transmission and intensity still remains high", should go into limitations.

 Response-We thank the reviewer for pointing out this comment. This section has been edited appropriately to the limitations section 

3- The author states that they are healthy patients, "A qualified physician attached to the project performed physical assessment of all the participants and only healthy participants were recruited", however it is necessary to mention which criteria this doctor used to consider a healthy patient.

 Response-We thank the reviewer for this comment. This has been additionally mentioned in lines 181 to 186 of the tracked manuscript and 107 to 112 of the clean manuscritpt

 Additional explanation for the reviewer is as below:

 During the primary vaccine study, the children who were enrolled and vaccinated in the study had all their physical examination done. Thus this data set being drawn from the children enrolled in the primary study had normal physical examination. Besides this was confirmed during extraction and verified by the principal investigator before being used in the analysis. 

4- Make it clear if the sample is random or if it was obtained for convenience. As part of a database already available, how was the recruitment of these patients? Was it random or not? Explain.

 Response- We thank the reviewer for this comment. As part of the data for the vaccine study and where physical examination was done, data for healthy infants as per the inclusion criteria were used. Wasn’t random because the data had to fit the pre-determined criteria where data set for participants in the database were extracted and verified before analysis. 

4 - The statistical analysis is robust and is based on non-parametric data. Explain which analyzes were performed to consider the data as non-parametric. Ideally, they should be parametric as there are many participants. Not being parametric, it shows that the values are not well distributed around an average. Shows that the data does not have a normal curve. In this case, put that in the LIMITATIONS.

 We thank the reviewer for this comment- in statistical analysis, before considerations were made, Shapiro-Wilk and Kolmogorov Smirnov test of normality were used in evaluating the data distribution and verified. This was verified as not normally distributed thus non-parametric tests were used and thus analysis also done in accordance with CLSI guideline in evaluating reference ranges . 

5- I suggest showing a histogram with the age distribution by age group. We thank you for this comment. This has been determined and re-verified as non-normal distribution

6- In results, explain the reason for the exclusion of 122 patients.

 Response-We thank the reviewer for this comment, this has been explained, also quoted as follows;

 ‘’ Reasons for exclusion included: Sickle cell anemia (N=2; 0.11%), fever (N=10; 0.57%), HIV exposed (N=101, 5.76%) and other acute illnesses (N=9; 0.51%)’’. 

7- Present a table with the reference intervals that can be easily used by academics and health professionals. In this table there is no need for a "p" value. Display only the parameter and its reference range. This table will be the author's main contribution to the scientific community.

 Response-We thank the reviewer for pointing out this comment .This has been rectified as per the comment and p values have been removed.________________________________________ 6. PLOS authors have the option to publish the peer review history of their article (what does this mean?). If published, this will include your full peer review and any attached files.

Do you want your identity to be public for this peer review? For information about this choice, including consent withdrawal, please see our Privacy Policy.

Reviewer #1: No

Reviewer #2: Yes: Eduardo Rodrigues Alves Junior

---

## [Editor Report · Decision Letter 1]

17 Dec 2020

Clinical Laboratory Hematology Reference Values among Infants Aged 1month to 17months in Kombewa Sub-County, Kisumu: A cross sectional study of rural population in Western Kenya

PONE-D-19-28401R1

Dear Dr. Ouma,

We’re pleased to inform you that your manuscript has been judged scientifically suitable for publication and will be formally accepted for publication once it meets all outstanding technical requirements.

Kind regards,

Stanley J. Robboy, MD

Academic Editor

PLOS ONE

Additional Editor Comments (optional):

None:
---

## [Editor Report · Acceptance letter]

8 Mar 2021

PONE-D-19-28401R1 

Clinical Laboratory Hematology Reference Values among Infants Aged 1month to 17months in Kombewa Sub-County, Kisumu: A cross sectional study of rural population in Western Kenya 

Dear Dr. Ouma:

I'm pleased to inform you that your manuscript has been deemed suitable for publication in PLOS ONE. Congratulations! Your manuscript is now with our production department. 

Kind regards, 

on behalf of

Dr. Stanley J. Robboy 

Academic Editor

PLOS ONE